# Sperm Behavior and Response to Melatonin under Capacitating Conditions in Three Sheep Breeds Subject to the Equatorial Photoperiod

**DOI:** 10.3390/ani11061828

**Published:** 2021-06-18

**Authors:** Melissa Carvajal-Serna, Jaime Antonio Cardozo-Cerquera, Henry Alberto Grajales-Lombana, Adriana Casao, Rosaura Pérez-Pe

**Affiliations:** 1Grupo BIOFITER (Biología, Fisiología y Tecnologías de la Reproducción), Departamento de Bioquímica y Biología Molecular y Celular, Instituto Universitario de Investigación en Ciencias Ambientales de Aragón (IUCA), Facultad de Veterinaria, Universidad de Zaragoza, 50013 Zaragoza, Spain; melissac@unizar.es (M.C.-S.); adriana@unizar.es (A.C.); 2Researcher Center Tibaitatá, AGROSAVIA (Corporación Colombiana de Investigación Agropecuaria), Bogotá 250047, Colombia; jcardozo@agrosavia.co; 3Departamento de Producción Animal, Facultad de Medicina Veterinaria y de Zootecnia, Universidad Nacional de Colombia, Bogotá 111321, Colombia; hagrajalesl@unal.edu.co

**Keywords:** ram sperm, equatorial, photoperiod, capacitation and melatonin

## Abstract

**Simple Summary:**

In temperate regions, sheep demonstrate seasonal reproduction regulated by changes in photoperiod. This regulation is mediated by nocturnal melatonin secretion. However, in equatorial regions, with no photoperiodic changes, sheep tend to breed in all seasons of the year. Despite this, changes in seminal composition or sperm quality have been reported throughout the year. We demonstrated that melatonin concentration in seminal plasma varies between rainy and dry seasons in three Colombian breeds (Colombian Creole, Romney Marsh, and Hampshire). As melatonin can exert direct effects on ram spermatozoa, in this study we hypothesized that melatonin could modulate sperm capacitation in equatorial-located breeds as we had previously reported in seasonal breeds from temperate regions. First, we assayed two media for in vitro capacitation and found that the increment in capacitated and acrosome-reacted sperm was higher in the so-called “cocktail medium” for the three breeds and in rainy and dry seasons. The addition of melatonin to the cocktail medium partially prevented the increase in capacitated spermatozoa from all breeds and during all seasons. This study could help in understanding how melatonin affects ram reproduction in the equatorial photoperiod.

**Abstract:**

In this study, we demonstrated that, in seasonal Mediterranean ovine breeds, supplementing the TALP medium with cAMP-elevating agents (the cocktail medium) is effective for achieving ram sperm capacitation, and that melatonin is able to regulate this phenomenon. We investigated the behavior under capacitating conditions using the TALP and cocktail mediums, and the response to melatonin, of spermatozoa from three sheep breeds (Colombian Creole, Romney Marsh, and Hampshire) subject to the equatorial photoperiod, during the dry and the rainy seasons. The cocktail medium was able to induce sperm capacitation, assayed by chlortetracycline staining and phosphotyrosine levels, to a greater extent than TALP, without a higher loss of viability (membrane integrity and viable spermatozoa without phosphatidylserine (PS) translocation). The addition of melatonin at 100 pM or 1 µM in the cocktail medium partially prevented the decrease in viability without PS translocation and the increase in capacitated spermatozoa from all breeds, with no significant effect on phosphotyrosine levels. Differences between breeds and seasons were evidenced. This study shows that melatonin is able to exert direct effects on spermatozoa in ovine breeds under equatorial photoperiod conditions, as it does in seasonal breeds located in temperate regions.

## 1. Introduction

Sperm capacitation in mammalians is a pre-requirement for fertilization and it occurs physiologically in the female reproductive tract, in the vicinity of the oocyte [1,2]. Sperm capacitation can be carried out in vitro, by adding substances that trigger the cAMP/PKA pathway [3,4]. Most spermatozoa from domestic species are easily capacitated in vitro in the presence of substances such as calcium, bicarbonate, and cholesterol acceptors [5,6,7,8]. However, ram spermatozoa are very difficult to capacitate in vitro and also require other substances that increase intracellular cAMP or avoid its degradation [4]. Our group previously demonstrated that a cocktail of substances added to the TALP medium successfully promotes capacitation of spermatozoa from Rasa Aragonesa [4,9], a Mediterranean breed raised in temperate latitudes with seasonal reproductive behavior. In temperate regions (>30° and <45° north or south latitude), sheep demonstrate seasonal reproduction, regulated mainly by melatonin according to changes in photoperiod (reviewed in [10]). Melatonin is secreted during the night by the pineal gland and displays a circadian rhythm with a nocturnal maximum and diurnal basal levels [11]. Sheep are short-day breeders, which means that higher levels of nocturnal melatonin secretion during autumn and winter nights in temperate regions have a stimulatory effect on their reproduction [12,13,14]. Although this seasonality is less marked in the ram than in the ewe [10,15], variations in testicular volume, sexual behavior, and semen quality have been detected between reproductive and non-reproductive seasons [16,17,18]. In the equatorial region (between 10° north and 10° south), with no changes in day length during the year, there are no changes in nocturnal melatonin levels acting as a cue for the timing of breeding. Thus, ovine breeds subject to equatorial photoperiod conditions tend to breed all-year-round [19]. Nevertheless, changes in seminal composition or sperm quality have also been reported in small ruminants throughout the year at this latitude, which could be attributed to environmental factors other than the photoperiod, such as rainfall [20,21,22].

However, melatonin can influence sheep reproduction through additional mechanisms other than photoperiod translation. This hormone can also be synthesized in the ram testes [23], it is present in seminal plasma and the female reproductive tract, and it can exert direct effects on ram spermatozoa by binding to specific membrane receptors (MT_1_ and MT_2_) [24]. Our group has demonstrated that melatonin is able to protect spermatozoa from Mediterranean seasonal rams against oxidative damage and apoptosis [25,26] and modulate sperm capacitation under in vitro conditions [25,27]. In a medium with a cocktail of cAMP-elevating agents, melatonin at micromolar concentrations was able to prevent sperm capacitation, whereas at lower concentrations it modified motile sperm subpopulations [28].

Despite the understanding of melatonin’s role in ram reproduction in temperate regions, little is known about this hormone’s functions in males under the equatorial photoperiod. Colombia, located at 4.5° N, has a 12L:12D photoperiod and an isothermal climate with a bimodal precipitation pattern, with two rainy and two dry seasons. Among ovine breeds in the country, Wool Creole, Romney Marsh, and Hampshire are predominant in high altitude areas [29]. Wool Creole is a native breed that has emerged from extensive crossbreeding since the 16th century [30]. This breed, adapted to the local environmental conditions, represents a vital genetic resource for small-scale agriculture in Andean regions [31]. As an alternative to this native breed, imported breeds such as Romney Marsh and Hampshire have been introduced since 1963. These breeds have a higher productive performance but are less adapted than the native breeds in terms of fertility [32].

In a previous study with these three Colombian breeds under equatorial photoperiod conditions, we observed differences in the melatonin concentration in seminal plasma obtained in the rainy and dry seasons [33,34]. As there was no change in night length that would modify the nocturnal secretion of melatonin, we hypothesized that these differences between seasons could be attributed to the content of phytomelatonin in the pasture [33]. Moreover, our group also demonstrated that spermatozoa from these Colombian rams also contain MT_1_ and MT_2_ receptors, and there are differences between breeds and seasons in the locations and densities of these receptors [34]. However, no previous studies have investigated how spermatozoa from these breeds, located in equatorial latitudes, respond under in vitro capacitating conditions, and whether exogenous melatonin could modulate this process.

Thus, the first objective of this study was to evaluate the response to in vitro capacitation in spermatozoa obtained from one native (Wool Creole) and two imported (Hampshire and Romney Marsh) sheep breeds reared in Colombia under a photoperiodic regimen of 12L:12D. The second objective was to elucidate whether melatonin can regulate ram sperm capacitation in these breeds in a medium with cAMP-elevating agents. Both evaluations were conducted during the rainy and dry seasons. This study could help in understanding how melatonin affects ram reproduction in the equatorial photoperiod.

## 2. Materials and Methods

### 2.1. Animals

Semen was obtained from twelve mature rams (2–5 years old) of three different sheep breeds (Wool Creole, Romney Marsh, and Hampshire; four rams of each breed). The animals were kept under uniform nutritional conditions at the Center for Ovine Research, Technological Development and Extension of the National University of Colombia, located in Mosquera (4°40′57″ N. 74°12′50″ W) at 2510 m above sea level. The rams’ diet was based on pasture (*Pennisetum clandestinum, Lolium perenne*), supplemented with 200 g of pellets (Leche Standard 70) and 15 g of mineralized salt (Universal F), both from FINCA S.A. The rams were kept under natural photoperiod conditions. The local amplitude of the photophase throughout the year fluctuates from 12 h 21′ (11 h 39′ of darkness) in the summer solstice to 11 h 49′ (12 h 11′ of darkness) in the winter solstice; i.e., with a total of 32′ of difference between the longest and the shortest days of the year. The climate of the region is classified as Cfb according to the Köppen Climate Classification System. The medium temperature is 13.6 °C, the annual variation between the coldest and hottest months being 0.7 °C. The daily temperature and relative air humidity varies from 18 °C to 7 °C and from 92% to 70%, respectively. The mean annual rainfall is 960 mm, with a mean of 205 rainy days per year. The experiments were performed in the rainy season (April–May) and dry season (June–July) based on precipitation data from the Institute of Hydrology, Meteorology, and Environmental Studies (IDEAM).

### 2.2. Semen Collection and Processing

Semen was collected once a week with the aid of an artificial vagina during four weeks in the rainy season and four weeks in the dry season. All procedures were performed in accordance with the Colombian Animal Protection Regulations (Law 84/1989, modified by Law 1774/2016) and under approval of the Bioethics Committee of the Faculty of Veterinary Medicine and Zootechnics of Bogotá, National University of Colombia (Project license: CB-074-2014). Before including rams in the study, individual ejaculates of each ram were analyzed separately during several months. All ejaculates showed ≥70% sperm motility (evaluated by an IVOS II CASA system; Hamilton Thorne, Beverly, MA, USA) and ≥75% normal sperm morphology. For the experiments described in this study, ejaculates from rams of the same breed were pooled and processed together in order to eliminate individual differences, [35].

After semen collection, the ejaculates were kept at 37 °C upon arrival at the laboratory located at the Tibaitatá research center which belongs to the Colombian Corporation for Agricultural Research (AGROSAVIA).

A seminal plasma-free sperm population was obtained using a dextran/swim-up procedure based on the modification proposed by Garcia-Lopez et al. 1996 [36]. It was performed in a swim-up medium (SM) devoid of NaHCO_3_ and CaCl_2_ [37] with the following composition: 200 mM sucrose, 50 mM NaCl, 18.6 mM sodium lactate, 21 mM HEPES, 10 mM KCL, 2.8 mM glucose, 0.4 mM MgSO_4_, 0.3 mM sodium pyruvate, 0.3 mM K_2_HPO_4_, and 5 mg/mL de BSA (pH adjusted to 6.5).

### 2.3. In Vitro Sperm Capacitation

Swim-up-selected spermatozoa (1.6 × 10^8^ cells/mL) were incubated in a humidified incubator for 3 h at 39 °C and with 5% CO_2_ in the air. Incubations were performed in a complete TALP medium [4,38] composed of 100 mM NaCl, 25 mM NaHCO_3_, 21.6 mM Na lactate, 10 mM HEPES, 3.1 mM KCl, 2 mM CaCl_2_, 1 mM Na pyruvate, 0.4 mM MgCl_2_, and 0.3 mM NaH_2_PO_4_ with 5 mM glucose, 5 mg/mL bovine serum albumin (BSA) and a pH of 7.2. As ram spermatozoa are difficult to capacitate in vitro, we also evaluated the addition of a cocktail of agents that increase intracellular cAMP to the TALP medium [4,7,9]. The cAMP-elevating compounds were 1 mM dibutyryl (dB)-cAMP, 1 mM caffeine, 1 mM theophylline, 0.2 mM okadaic acid, and 2.5 mM methyl-b-cyclodextrin (Sigma-Aldrich, Merck KGaA, St. Louis, MO, USA). We named this high cAMP medium the “cocktail medium”. This medium has already been proven for capacitating ram spermatozoa in certain seasonal breeds located in temperate regions but not in non-seasonal ones located in equatorial regions.

Melatonin was solubilized in DMSO (dimethyl sulphoxide) and PBS (phosphate-buffered saline: 137 mM NaCl, 8.1 mM Na_2_HPO_4_, 2.7 mM KCl, and 1.76 KH_2_PO_4_; pH 7.4) and then added to samples in the cocktail medium at a final concentration of 100 pM or 1 µM. The final concentration of DMSO in all the melatonin samples was 0.1%. To take into consideration the potential adverse effect of DMSO, the same concentration was included in cocktail- capacitated samples to which no melatonin had been added.

Thus, five experimental groups were analyzed in the present study: swim-up (spermatozoa selected by the dextran/swim-up procedure before inducing in vitro capacitation), cap-TALP (swim-up-selected spermatozoa incubated under capacitating conditions in TALP medium), cap-CK (swim-up-selected spermatozoa incubated under capacitating conditions in cocktail medium), and cap-CK-100 pM Mel and cap-CK-1 µM Mel (swim-up-selected spermatozoa incubated under capacitating conditions in cocktail medium in the presence of 100 pM and 1 µM melatonin, respectively).

### 2.4. Evaluation of Motility and Plasma Membrane Integrity

Motility and viability, assayed as plasma membrane integrity, were sequentially analyzed using an IVOS II CASA system (Hamilton Thorne, Beverly, MA, USA) on the same five fields under phase contrast and fluorescent illumination (using the Viadent filter), respectively [39]. For viability analysis, samples were previously stained using the VIADENT™ stain biz-benzamide trihydrochloride (Hoeschst 33258, 5 μg/mL; Hamilton Thorne, Beverly, MA, USA) which penetrates only in cells with a damaged membrane and attaches to the DNA, emitting fluorescence.

### 2.5. Detection of Membrane Phosphatidylserine Translocation

Annexin V (AnnV) is a protein with a high affinity for phosphatidylserine (PS). PS translocation is a marker of apoptotic-like changes in spermatozoa [40]. Simultaneous staining with 6-CFDA and Annexin V-Cy3.18 (Apoptosis Detection Kit; Sigma-Aldrich, St. Louis, MO, USA) was used in order to differentiate between viable spermatozoa with or without PS translocation and non-viable cells. The non-fluorescent 6-CFDA enters the cell and is converted to the green fluorescent compound 6-carboxyfluorescein (6-CF) only in viable cells, whereas red fluorescence can be observed only in cells with translocation of PS (AnnV+).

Aliquots of 300 µL (6 × 10^6^ cells diluted with 1 x binding buffer) were stained with 5 µL 6-CFDA (1 mM in DMSO) and 2 µL Annexin V-Cy3.18. Viable cells (6-CFDA+) were visualized in green with a standard fluorescein (Nikon B-2A) filter and AnnV+ cells (labeling PS exposure, Annexin V-Cy3.18+) in red with a rhodamine (Nikon G-2A) filter under an epifluorescence microscope (1000× magnification). At least 200 spermatozoa were counted per slide.

### 2.6. Assessment of Capacitation Status by Chlortetracycline (CTC) Staining

The capacitation status was determined by a modified chlortetracycline (CTC) fluorescence assay [41]. A CTC solution was prepared daily, adding 750 µM of CTC to a buffer composed of 130 mM NaCl, 20 mM Tris, and 5 µM cysteine (pH 7.8), which was then filtered through a 0.22 mm filter (Merck Millipore, Darmstadt, Germany). For a 20 µL sample (1.6 × 10^8^ cells/mL), 20 µL of CTC solution and 5 µL (12.2% *w*/*v*) paraformaldehyde (prepared in 0.5 M Tris-HCl, pH 7.8) were added. For evaluating CTC patterns, the samples were examined using a Nikon Eclipse E-200 microscope (Nikon Instruments, Kanagawa, Japan) under epifluorescence illumination in the V-2A filter (excitation filter 380–425 nm) at 1000× magnification. At least 200 spermatozoa per slide were classified into three categories [42]: non-capacitated (NC, with uniform fluorescence on the head, with or without a bright equatorial band), capacitated (C, showing fluorescence in the anterior region of the head), and acrosome-reacted (AR, showing no fluorescence on the head) spermatozoa.

### 2.7. Tyrosine Phosphorylation as Capacitation Assay

Sperm samples were subjected to protein membrane extraction. Proteins were obtained by suspending sperm (1.6 × 10^8^ cells/mL in 200 µL) in 200 µL of extraction sample buffer (ESB; composed of 2% SDS (sodium dodecyl sulfate-polyacrylamide (*w*/*v*) and 0.0626 M Tris-HCl, pH 6.8) [4]. The samples were incubated for 5 min at 100 °C and then centrifuged at 7500× *g* for 5 min at 4 °C. After recovering the supernatant, a mix of phosphatase and protease (10% *v*/*v*) inhibitors (Sigma-Aldrich, St. Louis, MO, USA), β-mercaptoethanol (5% *v*/*v*), glycerol (1% *v*/*v*), and bromophenol blue (0.002% (*v*/*v*) in 10% glycerol) were added and it was then stored at −20 °C until its use.

For sodium dodecyl-sulfate-polyacrylamide gel electrophoresis (SDS-PAGE), 15 μL of samples were loaded on 10% (*w*/*v*) SDS-PAGE gels for detection of phosphorylated protein in tyrosine residues. Proteins were separated by standard SDS-PAGE [43] and transferred onto polyvinylidene difluoride (PVDF) membrane using a transfer unit (Trans-Blot pack and Trans-Blot Turbo Transfer System, respectively, both from Bio-Rad Laboratories, Hercules, CA, USA).

Non-specific sites on the membrane were blocked with 5% BSA in PBS (*w*/*v*) for 4 h and membranes were incubated overnight at 4 °C with the mouse monoclonal anti-phosphotyrosine primary antibody (Clone 4G10; Millipore, Temecula, CA; Cat# 05–321, RRID: AB_309678), 1:1000 (*v*/*v*) in 0.1% (*w*/*v*) Tween-20–PBS containing 1% BSA (*w*/*v*). Additionally, a rabbit anti-actin antibody (Sigma-Aldrich, St. Louis, MO, USA; Cat# A2066, RRID: AB_476693), diluted 1:1000 (*v*/*v)* was used at the same time as a loading control. After three 15 min washes with 0.1% (*w*/*v*) Tween-20–PBS, membranes were incubated with a secondary donkey anti-rabbit IRDye 680-CW (LI-COR Biosciences; Cat# 926–32,223, RRID: AB_621845) and donkey anti-mouse IRDye 800-CW (LI-COR Biosciences, Lincoln, NE, USA; Cat# 926-32213, RRID: AB_621848) conjugated antibodies, both diluted 1:15,000 (*v*/*v*) in 0.1% (*w*/*v*) Tween-20–PBS containing 1% BSA (*w*/*v*) for 1 h at room temperature.

After extensive washing with 0.1% (*w*/*v*) Tween-20–PBS, membranes were scanned, and the intensity of the bands was measured with the Odyssey CLx Imaging System (LI-COR Biosciences, Lincoln, NE, USA). For densitometric evaluation, the signals corresponding to the high (60–250 kDa) and low (10–45 kDa) molecular weight phosphotyrosine proteins were considered, and the middle bands corresponding to the BSA signal were omitted.

### 2.8. Statistical Analysis

The obtained results were presented as means ± S.E.M. The number of replicates was four for all analyses (*n* = 4), except for PS translocation (*n* = 3). To determine whether there were significant differences in protein tyrosine phosphorylation between the treatments (swim-up, cap-TALP, cap-CK, cap-CK + 100 pM Mel, and cap-CK + 1 µM Mel), breeds, or seasons, two-way ANOVA tests followed by Bonferroni post hoc tests were used after the normality of the data was evaluated by the Kolmogorov–Smirnov test (Graph-Pad InStat software 3.01; San Diego, VA, USA). The percentage of total and progressive motility, viability, viable sperm without PS translocation, and CTC staining patterns were compared by means of Pearson’s chi-square test using SPSS software version 24.0, (IBM Corp, Armonk, NY, USA). *p* < 0.05 was used to indicate significant differences.

## 3. Results

### 3.1. Evaluation of the Changes after In Vitro Capacitation in Spermatozoa from Different Breeds and Seasons

#### 3.1.1. Changes in Motility after In Vitro Capacitation

Creole rams generally presented better total and progressive motility than the other two breeds in both seasons (*p* < 0.05, Table 1). After 3 h of incubation in capacitating conditions, the percentages of total and progressive motility decreased (*p* < 0.05) in relation to swim-up, and mainly when incubation was in the cocktail medium, for all breeds and seasons, except for the Creole breed during the rainy season. During this season, the percentages of total and progressive motility after in vitro capacitation remained high in the Creole breed and decreased significantly in the Hampshire breed (*p* < 0.05). The decline in motility was more pronounced during the dry season except for Hampshire spermatozoa (Table 1).

#### 3.1.2. Changes in Plasma Membrane after In Vitro Capacitation

The percentages of sperm viability (membrane integrity) in swim-up, cap-TALP, and cap-CK samples were higher in the rainy than in the dry seasons for all breeds (*p* < 0.05) except for Romney Marsh swim-up samples (Table 2). A significant decrease (*p* < 0.05) was observed after 3 h incubation in capacitating conditions both in TALP and cocktail media, with a difference between media for Creole and Hampshire breeds in the rainy season. Moreover, during both seasons the sperm viability after in vitro capacitation, with either media, was higher in the Creole breed than in the other breeds.

When PS translocation was evaluated simultaneously with plasma membrane integrity, significant differences between breeds were also revealed (Table 2). Swim-up samples from Creole rams showed a higher percentage of spermatozoa without PS translocation than those from the other two breeds. However, incubation in capacitating conditions significantly affected the Creole sperm in both seasons (*p* < 0.05), unlike the other two breeds that were affected only in the dry season (Romney Marsh) or not affected (Hampshire). Nonetheless, the rate of viable sperm without PS translocation remained higher than in the other breeds after the incubation in capacitating conditions in the Creole breed, especially when compared with the Hampshire breed (*p* < 0.05). On the other hand, there were no significant differences between seasons in any breed, except for cocktail samples from Hampshire rams, in which this parameter decreased even more during the dry season.

#### 3.1.3. Changes in Capacitation Status after In Vitro Capacitation

Swim-up samples from Hampshire rams presented much higher (*p* < 0.05) percentages of capacitated (35.5 ± 4.74 vs. 18.56 ± 2.72 and 19.75 ± 3.75 for Creole and Romney Marsh, respectively) and acrosome-reacted (6.00 ± 1.47 vs. 1.98 ± 0.57 and 1.00 ± 0.70 for Creole and Romney Marsh, respectively) spermatozoa in the rainy season than the other two breeds (Figure 1).

The incubation in TALP medium under capacitating conditions led to a significant decrease in the percentage of non-capacitated spermatozoa in both seasons for all three breeds (Figure 1). This decrease was concomitant with an increase in the rate of capacitated spermatozoa, except for Romney Marsh and Hampshire in the rainy season, and with an increment in acrosome reacted spermatozoa in most of the experimental groups. These changes were more evident when incubations were performed in the cocktail medium in both seasons for the three breeds. Moreover, during the dry season, a greater difference between the effect of the TALP and cocktail medium on the rate of capacitated spermatozoa was observed. Also, no differences between breeds were observed during the dry season when spermatozoa were incubated with the cocktail medium.

The more noteworthy dissimilarities were found when the effect of the season was analyzed. We found a substantial difference between dry and rainy seasons (*p* < 0.05) in the Romney Marsh and Hampshire spermatozoa response to the high cAMP medium, but not for the Creole spermatozoa. However, the Creole spermatozoa showed a higher response in the TALP medium during the rainy season (*p* < 0.05).

#### 3.1.4. Changes in Phosphorylation in Tyrosine Residues after In Vitro Capacitation

Incubation in the TALP medium did not significantly increase the phosphotyrosine signal compared to the swim-up samples, except for Creole in the rainy season (Figure 2). However, when the incubation was performed in the cocktail medium, a significant increment (*p* < 0.05) in the signal was observed in all breeds and seasons except for Romney Marsh in the rainy season. This increase was much higher in the dry than in the rainy season for the three breeds (Creole: 2335.86 ± 331.94 vs. 7211.04 ± 617.25; RM: 1254.89 ± 203.88 vs. 4247.92 ± 677.23; HS: 2091.26 ± 332.26 vs. 5468.36 ± 325.20; rainy and dry season, respectively) (Figure 2), with significant differences between seasons (*p* < 0.05). The increment was lower in spermatozoa from Romney Marsh than in spermatozoa from the other two breeds.

### 3.2. Evaluation of the Effect of Melatonin on In Vitro Capacitation in Spermatozoa from Different Breeds and Seasons

#### 3.2.1. Effects on Motility during In Vitro Capacitation

When samples were incubated in capacitating conditions in the cocktail medium, the presence of melatonin had no effect on total motility in most experimental samples compared to the cap-CK without the hormone, except for a decrease in Creole and Hampshire spermatozoa at 100 pM in the rainy season and the dry season, respectively, and an increase in Romney Marsh spermatozoa at 1 µM in the rainy season (Table 3). Regarding progressive motility, there were also no significant effects, except for a slight decrease (*p* < 0.05) in Hampshire spermatozoa when melatonin was added at both concentrations in the dry season (Table 3).

#### 3.2.2. Effects on Plasma Membrane during In Vitro Capacitation

In the dry season, the presence of melatonin in cap-CK samples did not affect the membrane integrity (viability). In the rainy season, 1 µM of melatonin (*p* < 0.05) had a positive effect in Romney Marsh spermatozoa and a negative one in Hampshire ones, whereas 100 pM had a slightly negative effect in the Creole breed compared to cap-CK without the hormone (Table 4), in concordance with its impact on motility in these breeds (Table 3).

However, when the percentage of intact spermatozoa without PS translocation was evaluated, the effect of melatonin was positive in all breeds in both seasons, except in the Creole breed during the dry season. The effective melatonin concentration depended on the breed and season. During the rainy season, an increase in this parameter in Creole and Hampshire spermatozoa was observed with 100 pM melatonin. In the Romney Marsh breed, 1 µM melatonin was the effective concentration (*p* < 0.05) in both seasons (Table 4).

#### 3.2.3. Effects on Capacitation Status during In Vitro Capacitation

The addition of melatonin at both concentrations (100 pM and 1 µM) in the cocktail medium partially prevented the increase in capacitated sperm provoked by the incubation in capacitating conditions in all breeds (Figure 3). Thus, in the cap-CK samples with melatonin, a higher percentage of non-capacitated sperm (*p* < 0.05) was observed compared to the cap-CK samples without the hormone. This effect was concomitant with a lower rate of capacitated sperm (*p* < 0.05), except for Romney Marsh in the dry season. Although the preventive effect of melatonin on sperm capacitation was observed in the three breeds, it was more noticeable in the Romney Marsh and Hampshire rams during the rainy season. An impact on reacted spermatozoa was observed only in the rainy season; the Cap-CK samples with melatonin showed a lower percentage than the Cap-CK without the hormone. This effect was evident at both concentrations in the Hampshire breed and at 100 pM or 1 µM for the Creole and Romney Marsh breeds, respectively.

In general, the effect of melatonin was more evident in the rainy season than in the dry season for all breeds.

#### 3.2.4. Effects on Phosphorylation in Tyrosine Residues during In Vitro Capacitation

In general, the addition of melatonin to samples incubated under capacitating conditions had no significant effects on phosphotyrosine levels, except for an increase in Creole spermatozoa in the rainy season (Figure 4). Despite the decrease in the signal observed in the dry season in the Creole and Hampshire breeds, it was not significant when compared with Cap-CK without the hormone (Figure 4). Significant differences between seasons were observed (*p* < 0.05).

## 4. Discussion

In this study, we evaluated the response to in vitro capacitation in ram spermatozoa from three Colombian breeds located in the equatorial region under two different seasonal climatic conditions and in natural grazing. The evaluation of some sperm quality variables before capacitation revealed significant differences between the native breed (Creole) and the introduced ones (Romney Marsh and Hampshire). Spermatozoa from Creole rams showed higher total and progressive motility, better plasma membrane integrity, and less PS translocation than spermatozoa from the introduced breeds, in both rainy and dry seasons. When comparing between seasons, sperm quality variables were generally better in the rainy season for Creole and Romney Marsh and in the dry season for Hampshire.

After incubation in capacitating conditions, both in TALP and in TALP with a cocktail of agents that increased the cAMP (cocktail medium), most of these variables decreased, especially with the cocktail medium. These changes were more acute in the dry season, except for in the Hampshire breed. In the rainy season, total and progressive motility and viability in Creole spermatozoa remained higher after incubation in the cocktail medium than those of the imported breeds. Our previous results in Rasa Aragonesa rams in the Mediterranean region also showed a decrease in these variables but with no statistical differences between both media [28].

Regarding the capacitation status, incubation in both media led to a significant decrease in the percentages of non-capacitated spermatozoa in both seasons and for the three breeds. This decrease was significantly more acute when incubation was performed in the cocktail medium than in TALP, this being concomitant with increases in the rates of capacitated and acrosome-reacted spermatozoa. Changes in capacitation status would explain the decrease in motility, especially progressive motility, due to the sperm hyperactivation associated with capacitation [9,28,44]. The decrease in spermatozoa without PS translocation could also be attributed to capacitation. Translocation of phosphatidylserine (PS) from the inner to the outer leaflet of the plasma membrane has been defined as one of the earliest signs of apoptosis [45,46], but it is also associated with plasma membrane scrambling related to capacitation and the acrosome reaction [47]. Moreover, it is worth noting that capacitation and apoptosis share signal transduction pathways [48]. When comparing between breeds, it can be seen that spermatozoa from Hampshire rams obtained in the rainy season were significantly more capacitated before incubation than spermatozoa from the other breeds. This would explain the levels of capacitated sperm after incubation in the cocktail medium being significantly higher in this breed. However, in the dry season, no differences in the percentages of capacitated spermatozoa between breeds were observed either before or after incubation in the cocktail medium.

A significant increase in the phosphorylation in tyrosine residues of proteins from samples incubated in the cocktail medium was also observed in both seasons for all breeds, unlike what occurred when incubation was carried out in TALP. Previously, we demonstrated that protein tyrosine phosphorylation of membrane proteins is related to the capacitation state in ram spermatozoa [7,49]. Also, we showed that the presence of cAMP-elevating agents in the capacitation medium is necessary to achieve a high increase in the phosphorylation levels in Mediterranean seasonal rams [4,50]. In the present study, the effect of the cocktail medium was more evident in the dry season. However, this increase was much less marked in the Romney Marsh breed than in the other two breeds.

Therefore, the cocktail medium was able to induce sperm capacitation to a greater extent than TALP, without a higher loss of sperm quality, in these three breeds in both seasons. Consequently, we used the cocktail medium to test the ability of melatonin to exert direct effects on spermatozoa from breeds subjected to the equatorial period, as occurs in breeds of reproductive seasonality located in other latitudes [4,9,28,51].

In the present study, we assayed two concentrations of melatonin, 100 pM and 1 µM, because we have previously observed different effects on Rasa Aragonesa spermatozoa depending on its concentration. In this Mediterranean seasonal breed, melatonin at 100 pM had a positive impact on viability, whereas 1 µM provoked a slight decrease in progressive motility when added to the cocktail medium in capacitating conditions [28]. In Colombian breeds, the positive effect of 100 pM of melatonin was only observed in the Romney Marsh breed in the rainy season and the effect on progressive motility was evidenced in the Hampshire breed in the dry season at both concentrations of the hormone. The effect of melatonin was more evident when the percentage of capacitated spermatozoa was evaluated. The addition of melatonin at both concentrations in the cocktail medium partially prevented the increase in capacitated sperm provoked by the incubation in capacitating conditions in all breeds. Our group has previously shown that melatonin can regulate in vitro capacitation mainly via the MT_2_ receptor in sperm from temperate-located rams [27]. In a previous study [34], we demonstrated that both melatonin receptors, MT_1_ and MT_2_, are also present in the spermatozoa of these three Colombian breeds raised under equatorial photoperiodic conditions. In the present study, the effect of melatonin was, in general, more evident in the rainy season than in the dry season for all breeds. This could be attributed to previously reported differences in the melatonin concentration in seminal plasma between seasons, with higher levels in the dry season for the three studied breeds [33]. Thus, the previous exposure of spermatozoa to high endogenous melatonin levels during the dry season could mitigate the effect of this hormone when it is added in vitro. This effect has been previously reported in somatic cells, where the exposure to a high level of melatonin can desensitize melatonin receptors [52]. Differences between breeds could also be attributed to previously described differences in the density of melatonin receptors [34]. The lesser effect of melatonin on sperm capacitation in Creole rams could be due to a lower MT_2_ density in spermatozoa from this breed than the other two, as we described in a previous study [34]. Colombian Creole sheep are a native breed that, after five centuries of breeding in a 12L:12D photoperiod, are much more adapted to the equatorial climate and photoperiod than the Hampshire and Romney Marsh breeds, which were introduced into the country only 50 years ago. This adaptation to a non-seasonal photoperiod could be reflected in the reduction of the MT_2_ receptor density, but the fact that it has not disappeared further supports the idea that melatonin has other physiological functions than seasonal control, as we pointed out in our previous work [23,53]. In the studied Colombian breeds, both assayed melatonin concentrations partially prevented the increase in capacitated sperm, whereas in Rasa Aragonesa spermatozoa only the higher concentration was effective. However, when phosphotyrosine levels were analyzed, no significant effect of the hormone was observed, in contrast with the decrease evidenced in Rasa Aragonesa spermatozoa in the presence of 1 µM melatonin [28].

Melatonin also had a significant effect on the percentage of viable sperm without PS translocation, which remained high in capacitating conditions in all breeds and in both seasons, except in the Creole breed during the dry season. As mentioned above, PS translocation could be related to capacitation or apoptosis [48]. The decapacitating effect of melatonin could explain the lower levels of sperm with PS translocation, but an antiapoptic effect of melatonin has also been reported in somatic cells [49] as well as in spermatozoa [25,54]. This antiapoptotic action seems to be mediated by melatonin binding to its receptors, specifically MT_1_ [55,56]. The observed variations between seasons, the effect being more evident in the rainy than in the dry season, could equally be attributed to the higher endogenous levels of melatonin in seminal plasma in the dry season [33], and consequently to a desensitization of the melatonin receptors.

## 5. Conclusions

To sum up, the present study shows that incubation in a medium with cAMP-elevating agents effectively achieves in vitro capacitation in spermatozoa from different ram breeds under the equatorial photoperiod. However, the response to the in vitro capacitation was different between breeds and seasons. The addition of melatonin to the medium with cAMP-elevating agents partially prevented the increase in capacitated spermatozoa and the decrease in viable spermatozoa without PS translocation in the three studied breeds, with differences between breeds and seasons. This study shows that melatonin is able to exert direct effects on spermatozoa in ovine breeds located under the equatorial photoperiod, as it does in seasonal breeds located in temperate regions.

## Figures and Tables

**Figure 1 animals-11-01828-f001:**
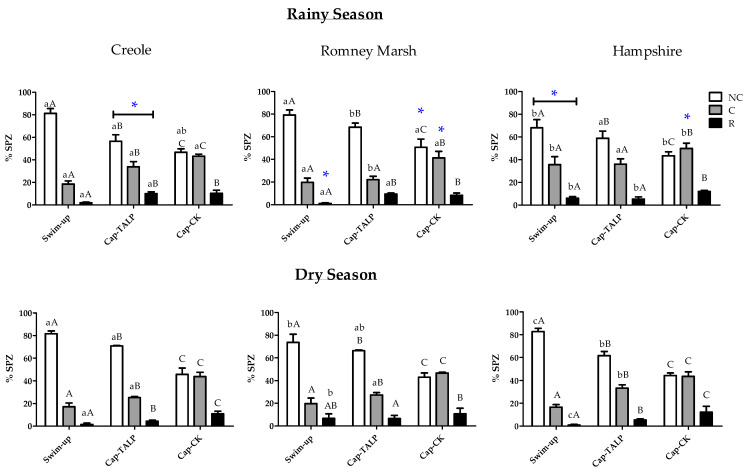
Assessment of capacitation status of ram spermatozoa, evaluated by chlortetracycline (CTC) staining, before (swim-up) and after 3 h incubation at 39 °C and 5% CO_2_ (capacitating conditions) in TALP (cap-TALP) or TALP with cAMP-elevating agents (cap-CK). The distributions were done by breed and season (rainy or dry). Data for percentages of non-capacitated (NC), capacitated (C), and acrosome-reacted (R) spermatozoa are expressed as means ± S.E.M. (*n* = 4). * represents significant differences (*p* < 0.05) between seasons within the same treatment and breed; different capital letters represent significant differences between treatments (swim-up, Cap-TALP, and Cap-CK) within a breed and season; different lowercase letters represent significant differences between breeds within a treatment and season.

**Figure 2 animals-11-01828-f002:**
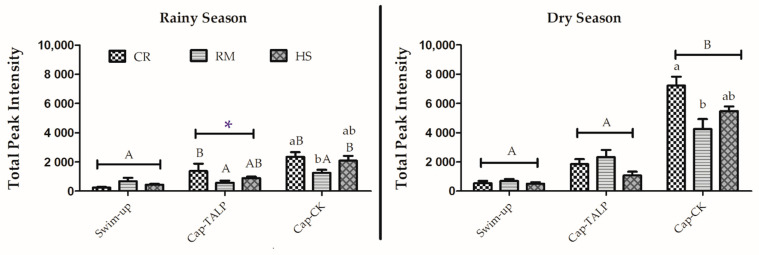
Protein tyrosine phosphorylation evaluated by densitometry in samples before (swim-up) and after 3 h of incubation at 39 °C and 5% CO_2_ (capacitating conditions) in TALP (cap-TALP) or TALP with cAMP-elevating agents (cap-CK). Data for the Creole (CR), Romney Marsh (RM), and Hampshire (HS) breeds are distributed by season (rainy or dry) and expressed as means ± S.E.M (*n* = 4). * represents significant differences (*p* < 0.05) between seasons within the same treatment and breed; different capital letters represent significant differences between treatments (swim-up, Cap-TALP, and Cap-CK) within a breed and season; different lowercase letters represent significant differences between breeds within a treatment and season.

**Figure 3 animals-11-01828-f003:**
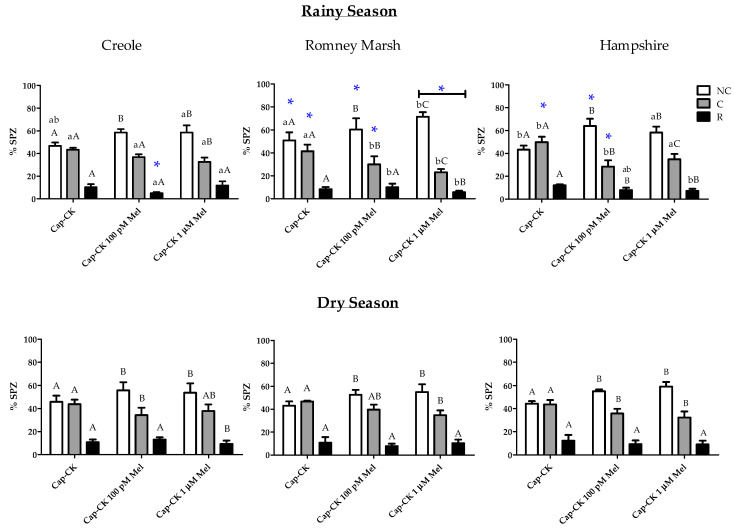
Assessment of capacitation status of ram spermatozoa, evaluated by chlortetracycline (CTC) staining, after 3 h incubation at 39 °C and 5% CO_2_ (capacitating conditions) in TALP with cAMP-elevating agents without (cap-CK) or with 100 pM and 1 µM melatonin (cap-CK-100 pM MEL and Cap-CK-1 µM MEL). The distributions were done by breed and season (rainy or dry). Data for percentages of non-capacitated (NC), capacitated (C), and acrosome-reacted (R) spermatozoa are expressed as means ± S.E.M. (*n* = 4). * represents significant differences (*p* < 0.05) between seasons within the same treatment and breed; different capital letters represent significant differences between treatments (Cap-CK, Cap-CK-100 pM MEL and Cap-CK-1 µM MEL) within a breed and season; different lowercase letters represent significant differences between breeds within a treatment and season.

**Figure 4 animals-11-01828-f004:**
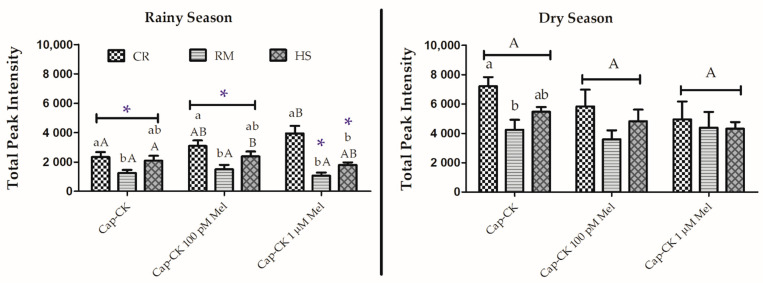
Protein tyrosine phosphorylation evaluated by densitometry in samples after 3 h of incubation at 39 °C and 5% CO_2_ (capacitating conditions) in TALP with cAMP-elevating agents without (cap-CK) or with 100 pM and 1 µM melatonin (Cap-CK-100 pM MEL and Cap-CK-1 µM MEL). Data for the Creole (CR), Romney Marsh (RM), and Hampshire (HS) breeds are distributed by season (rainy or dry) and expressed as means ± S.E.M (*n* = 4). * represents significant differences (*p* < 0.05) between seasons within the same treatment and breed; different capital letters represent significant differences between treatments (Cap-CK, Cap-CK-100 pM MEL, and Cap-CK-1 µM MEL) within a breed and season; different lowercase letters represent significant differences between breeds within a treatment and season.

**Table 1 animals-11-01828-t001:** Percentage of total and progressive sperm motility in the rainy and dry seasons in three ram breeds before (swim-up) and after 3 h of incubation at 39 °C and 5% CO_2_ (capacitating conditions) in TALP (cap-TALP) or TALP with cAMP-elevating agents (cap-CK).

Variables	Total Motility			Progressive Motility		
Breed	Creole	Romney Marsh	Hampshire	Creole	Romney Marsh	Hampshire
	**Rainy season**					
Swim-up	77.9 ± 3.7 ^aA^	72.8 ± 3.0 ^bA^	65.7 ± 3.9 *^cA^	64.7 ± 11.7 ^aA^	41.3 ± 9.0 *^bA^	43.9 ± 5.1 *^bA^
Cap-TALP	68.6 ± 3.0 *^aB^	55.4 ± 6.2 *^bB^	29.0 ± 5.2 *^cB^	44.4 ± 4.9 *^aB^	27.5 ± 5.9 *^bB^	5.8 ± 3.0 *^cB^
Cap-CK	76.7 ± 2.8 *^aA^	47.2 ± 6.8 *^bC^	22.0 ± 3.8 *^cC^	32.8 ± 7.0 *^aC^	17.0 ± 4.1 *^bC^	1.2 ± 0.3 *^cC^
	**Dry season**					
Swim-up	78.8 ± 3.4 ^aA^	69.2 ± 4.0 ^bA^	72.9 ± 3.3 ^bA^	59.8 ± 3.9 ^aA^	53.3 ± 5.0 ^bA^	54.2 ± 0.6 ^abA^
Cap-TALP	46.9 ± 13.6 ^aB^	39.8 ± 3.6 ^bB^	36.7 ± 9.6 ^bB^	23.0 ± 11.5 ^aB^	13.0 ± 4.6 ^bB^	13.2 ± 5.3 ^bB^
Cap-CK	33.8 ± 5.8 ^C^	33.8 ± 5.2 ^B^	28.1 ± 8.5 ^C^	5.7 ± 1.9 ^C^	3.9 ± 1.6 ^C^	5.9 ± 3.0 ^C^

Values are expressed as means ± S.E.M. (*n* = 4). * represents significant differences (*p* < 0.05) between seasons within the same treatment and breed; different lowercase letters in the same row represent significant differences between breeds within a treatment and season; different capital letters in the same column represent significant differences between treatments (swim-up, Cap-TALP, and Cap-CK) within a breed and season.

**Table 2 animals-11-01828-t002:** Percentages of viability (plasma membrane integrity) and viable sperm without PS translocation in the rainy and dry seasons in three ram breeds before (swim-up) and after 3 h of incubation at 39 °C and 5% CO_2_ (capacitating conditions) in TALP (cap-TALP) or TALP with cAMP-elevating agents (cap-CK).

Variables	Viability (Plasma Membrane Integrity %)			Viable Sperm without PS Translocation (%)		
Breed	Creole	Romney Marsh	Hampshire	Creole	Romney Marsh	Hampshire
	**Rainy season**					
Swim-up	88.6 ± 3.6 *^aA^	80.4 ± 5.3 ^bA^	86.5 ± 4.2 *^aA^	62.4 ± 5.2 ^aA^	47.0 ± 3.5 ^bA^	40.4 ± 9.2 ^bA^
Cap-TALP	80.1 ± 1.3 *^aB^	68.2 ± 5.9 *^bB^	49.1 ± 4.5 *^cB^	53.6 ± 5.9 ^aB^	53.0 ± 5.0 ^aA^	35.4 ± 0.7 ^bA^
Cap-CK	86.0 ± 1.0 *^aA^	63.0 ± 6.5 *^bB^	40.6 ± 3.9 *^cC^	56.3 ± 6.4 ^aAB^	55.5 ± 7.5 ^aA^	40.7 ± 1.3 *^bA^
	**Dry season**					
Swim-up	85.7 ± 2.9 ^aA^	79.2 ± 3.3 ^bA^	80.8 ± 3.9 ^abA^	70.4 ± 11.9 ^aA^	50.0 ± 6.5 ^bA^	39.8 ± 0.2 ^cA^
Cap-TALP	68.9 ± 8.5 ^aB^	54.6 ± 3.9 ^bB^	55.6 ± 2.2 ^bB^	70.0 ± 2.6 ^aA^	42.0 ± 10.6 ^bB^	37.0 ± 3.0 ^bA^
Cap-CK	71.7 ± 4.8 ^aB^	56.7 ± 3.9 ^bB^	53.5 ± 8.3 ^bB^	51.3 ± 12.0 ^aB^	43.0 ± 7.2 ^aAB^	32.5 ± 4.5 ^bA^

Values are expressed as means ± S.E.M. (*n* = 4 and *n* = 3 for viability and PS translocation, respectively). * represents significant differences (*p* < 0.05) between seasons within the same treatment and breed; different lowercase letters in the same row represent significant differences between breeds within a treatment and season; different capital letters in the same column represent significant differences between treatments (swim-up, Cap-TALP and Cap-CK) within a breed and season.

**Table 3 animals-11-01828-t003:** Percentages of total and progressive sperm motility in the rainy and dry seasons in three ram breeds after 3 h of incubation at 39 °C and 5% CO_2_ (capacitating conditions) in TALP with cAMP-elevating agents without (cap-CK) or with 100 pM and 1 µM melatonin (Cap-CK-100 pM MEL and Cap-CK-1 µM MEL).

Variables	Total Motility			Progressive Motility		
Breed	Creole	Romney Marsh	Hampshire	Creole	Romney Marsh	Hampshire
	**Rainy season**					
Cap-CK	76.7 ± 2.8 *^aA^	47.2 ± 6.8 *^bA^	22.0 ± 3.8 *^cA^	32.8 ± 7.0 *^aA^	17.0 ± 4.1 *^bA^	1.2 ± 0.3 *^cA^
Cap-CK-100 pM MEL	67.8 ± 5.1 *^aB^	51.0 ± 5.4 *^bA^	22.9 ± 3.9 ^cA^	30.0 ± 4.9 *^aA^	17.7 ± 4.1 *^bA^	2.3 ± 0.9 ^cA^
Cap-CK-1 µM MEL	77.5 ± 2.1 *^aA^	57.1 ± 2.4 *^bB^	18.6 ± 1.5 ^cA^	32.3 ± 2.5 *^aA^	18.1 ± 1.8 *^bA^	1.7 ± 0.6 ^cA^
	**Dry season**					
Cap-CK	33.8 ± 5.8 ^A^	33.8 ± 5.2 ^A^	28.1 ± 8.5 ^A^	5.7 ± 1.9 ^A^	3.9 ± 1.6 ^A^	5.9 ± 3.0 ^A^
Cap-CK-100 pM MEL	30.5 ± 5.6 ^aA^	33.9 ± 6.5 ^aA^	21.9 ± 5.0 ^bB^	5.1 ± 2.3 ^A^	3.8 ± 1.3 ^A^	2.4 ± 0.7 ^B^
Cap-CK-1 µM MEL	35.1 ± 6.6 ^aA^	30.0 ± 4.0 ^aA^	23.3 ± 6.1 ^bA^	5.9 ± 2.8 ^A^	3.6 ± 1.3 ^A^	3.0 ± 1.3 ^B^

Values are expressed as means ± S.E.M. (*n* = 4). * represents significant differences (*p* < 0.05) between seasons within the same treatment and breed; different lowercase letters in the same row represent significant differences between breeds within a treatment and season; different capital letters in the same column represent significant differences between treatments (Cap-CK, Cap-CK-100 pM MEL, and Cap-CK-1 µM MEL) within a breed and season.

**Table 4 animals-11-01828-t004:** Percentages of viability (plasma membrane integrity) and viable sperm without PS translocation in the rainy and dry seasons in three ram breeds after 3 h of incubation at 39 °C and 5% CO_2_ (capacitating conditions) in TALP with cAMP-elevating agents without (cap-CK) or with 100 pM and 1 µM melatonin (Cap-CK-100 pM MEL and Cap-CK-1 µM MEL).

Variables	Viability (Plasma Membrane Integrity %)			Viable Sperm without PS Translocation (%)		
Breed	Creole	Romney Marsh	Hampshire	Creole	Romney Marsh	Hampshire
	**Rainy season**					
Cap-CK	86.0 ± 1.0 *^aA^	63.0 ± 6.5 *^bA^	40.6 ± 3.9 *^cA^	56.3 ± 6.4 ^aA^	55.5 ± 7.5 ^aA^	40.7 ± 1.3 *^bA^
Cap-CK-100 pM MEL	80.3 ± 3.7 *^aB^	65.7 ± 7.1 *^bA^	44.0 ± 6.3 ^cA^	72.9 ± 6.4 *^aB^	57.0 ± 6.2 ^bA^	49.7 ± 7.5 *^bB^
Cap-CK-1 µM MEL	84.8 ± 1.6 *^aAB^	73.7 ± 3.5 *^bB^	36.1 ± 5.8 *^cB^	64.7 ± 3.3 *^aA^	62.7 ± 7.7 *^aB^	44.2 ± 8.4 ^bA^
	**Dry season**					
Cap-CK	71.7 ± 4.8 ^aA^	56.7 ± 3.9 ^bA^	53.5 ± 8.3 ^bA^	51.3 ± 12.0 ^aA^	43.0 ± 7.2 ^aA^	32.5 ± 4.5 ^bA^
Cap-CK-100 pM MEL	72.2 ± 4.4 ^aA^	54.7 ± 4.5 ^bA^	50.0 ± 5.7 ^bA^	51.7 ± 5.4 ^aA^	50.0 ± 1.5 ^aAB^	41.0 ± 4.0 ^bAB^
Cap-CK-1 µM MEL	71.5 ± 4.2 ^aA^	54.0 ± 2.60 ^bA^	50.1 ± 3.8 ^bA^	46.0 ± 4.6 ^aA^	55.7 ± 0.7 ^bB^	47.5 ± 10.5 ^aB^

Values are expressed as means ± S.E.M. (*n* = 4 and *n* = 3 for viability and PS translocation, respectively). * represents significant differences (*p* < 0.05) between seasons within the same treatment and breed; different lowercase letters in the same row represent significant differences between breeds within a treatment and season; different capital letters in the same column represent significant differences between treatments (Cap-CK, Cap-CK-100 pM MEL, and Cap-CK-1 µM MEL) within a breed and season.

## Data Availability

The datasets generated for this study can be found in the figshare repository 10.6084/m9.figshare.14528916.

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
