# Peer review of "Sperm Behavior and Response to Melatonin under Capacitating Conditions in Three Sheep Breeds Subject to the Equatorial Photoperiod"

_animals, 2021, doi:10.3390/ani11061828_

Round 1

Reviewer 1 Report

General comments: The authors have provided a generally well written manuscript that examines in vitro capacitation characteristics of spermatozoa from Creole, Romney Marsh, and Hampshire rams during both a rainy (April-May) season and a dry season (June-July) in Colombia with minimal variation in day length. Four rams of each breed were collected weekly (number of weeks of collection in each season need to be indicated in Methods to ascertain the number of observations) but semen was pooled within breed thereby eliminating the variation due to individual rams which would have been of interest to investigate. Following a swim-up procedure, 5 capacitation treatments included the swim-up control, TALP, CK, CK + 100 pM melatonin, and CK + 1 µM melatonin. Variables (not parameters) examined included total motility, progressive motility, plasma membrane integrity, viable sperm without PS translocation, capacitation status, and protein tyrosine phosphorylation. There are some areas of question as well as numerous minor areas of suggested edits. The manuscript was tricky to review because line numbers were not included for ease of identification location of needed changes.

Specific comments:

P1: “First” vs “Firstly” ?

P1: Hyphenate: “acrosome-activated”

P2: “… bimodal precipitation pattern, with two rainy and two dry seasons.”

P2: Use “extensive crossbreeding” rather than “multiple crossbreeding”

P3, L1 and P12, L2: Change “Since” to “Because”

P3, Intro, last para: “… under a photoperiodic regimen of 12L:12D.”

P3, Section 2.1: What type of pellets? Should be 200 g and 15 g, respectively.

P3, Section 2.2: How many weeks was semen collected in each season? How many total observations for each breed?

P3, Section 2.2: Observations are reduced by pooling semen and variation within a breed may be important. Were any precautions taken to ensure that semen from all 12 rams was potentially viable?

P6, Section 2.8: An analysis of variance might have included at model with breed, season, and a breed*season interaction? Is that how it was done? The results and discussion point out what I would expect to be significant breed*season interactions and could be reported accordingly.

Tables 1 and 2 and in multiple places in the Discussion: Use Variables” rather than “Parameters” The word parameter refers to population characteristics such as mean, standard error, etc.

Tables 1 and 2 and P8, 3rd line from the bottom (331.9): Use periods “.” rather than commas “,” to indicate decimal places. Also, means and SE in the tables and text probably could be rounded to one decimal place without loss of important information. This would make tables appear less cluttered. Table 1 also needs to indicate that values shown are percentages.

Table 2; Why are the number of observations lower (3 vs 4) for PS Translocation data? May need to indicate in the Methods.

For both tables and figures, the legends should more completely describe the treatments so that the reader does not have to refer back to the text to understand tables and figures independently.

Figure 2 legend: This sentence makes no sense: “High molecular weight proteins are represented above in the figure, while low molecular weight proteins are located below.” Please explain or delete.

Author Response

Thank you. Sorry for the missing line numbers. I can see them (the numbers) in my computer, and I hope they will be visible for you in the reviewed manuscript.

The number of weeks of collection in each season was 4. This information is now included in the manuscript (Line 146).

Specific comments:

- P1: “First” vs “Firstly” ?

Line 24, firstly was replaced by first.

- P1: Hyphenate: “acrosome-activated”

Line 25: acrosome-reacted has been hyphenated

- P2: “… bimodal precipitation pattern, with two rainy and two dry seasons.”

Line 92: regime has been replaced by pattern

- P2: Use “extensive crossbreeding” rather than “multiple crossbreeding”

Line 94: multiple has been replaced by extensive

- P3, L1 and P12, L2: Change “Since” to “Because”

Line 102 and 517: since has been replaced by because

- P3, Intro, last para: “… under a photoperiodic regimen of 12L:12D.”

Line 120: regime has been replaced by regimen

- P3, Section 2.1: What type of pellets? Should be 200 g and 15 g, respectively.

The pellets are a feed supplement with high protein levels named Leche Standar 70 from the branch FINCA S.A. This information is now included in the manuscript (Line 131).

- P3, Section 2.2: How many weeks was semen collected in each season? How many total observations for each breed?

Semen was collected once a week with the aid of an artificial vagina during four weeks in the rainy season and four weeks in the dry season (this information is now included in the manuscript, Line 146), so eight ejaculates of each animal were obtained. As ejaculates of the same breed were pooled together, eight observations for each breed were obtained, four in each season.  When we designed the experiments, we decided to limit the experimental work only in the weeks in the middle of each season to avoid inter-seasonal periods. In consequence, a few weeks in each season were available for experimental work and was not possible to get more replicates.

- P3, Section 2.2: Observations are reduced by pooling semen and variation within a breed may be important. Were any precautions taken to ensure that semen from all 12 rams was potentially viable?

In the Center for Ovine Research, only four Creole, four Romney Marsh, and five Hampshire rams were hosted and available for research.  Prior to the experiments, individual ejaculates of each ram were analyzed separately during several months. All ejaculates showed ≥70% sperm motility (evaluated by IVOSII CASA system; Hamilton Thorne, Beverly, MA, USA) and ≥ 75% normal sperm morphology, except for those from one Hampshire ram, that showed lower values and was not included in the study due to its low sperm quality. Finally, four rams of each breed with good sperm quality were included in this study. Part of this information is now included in the manuscript (Line 151).

- P6, Section 2.8: An analysis of variance might have included at model with breed, season, and a breed*season interaction? Is that how it was done? The results and discussion point out what I would expect to be significant breed*season interactions and could be reported accordingly.

Of all the variables analyzed in this study only protein tyrosine phosphorylation values were continuous data that were normally distributed. Thus, it was the only variable analyzed by ANOVA test, in which interaction between the studied parameters (breed, season and treatment) was also analyzed. The rest of the variables were categorical variables, and therefore, they cannot be analyzed by ANOVA test, so we used chi-square test instead.

ANOVA test demonstrated a breed*season interaction (apart from an effect of the treatment). After that, a Bonferroni post-hoc test was performed to identify the specific significant differences that are included in the Fig.2.

A more detailed explanation of the ANOVA model was included in the text (line 284).

- Tables 1 and 2 and in multiple places in the Discussion: Use “Variables” rather than “Parameters” The word parameter refers to population characteristics such as mean, standard error, etc.

The word “parameters” has been replaced by “variables” in tables and in discussion (lines 469, 473 and 477).

- Tables 1 and 2 and P8, 3rd line from the bottom (331.9): Use periods “.” rather than commas “,” to indicate decimal places. Also, means and SE in the tables and text probably could be rounded to one decimal place without loss of important information. This would make tables appear less cluttered. Table 1 also needs to indicate that values shown are percentages.

Sorry for the mistake. Commas has now been replaced by periods in the tables and in the text (line 390). We have rounded to one decimal and have indicate that values are percentages.

-Table 2; Why are the number of observations lower (3 vs 4) for PS Translocation data? May need to indicate in the Methods.

We had a trouble with the processing of some samples, and we only were able to analyze three samples in each season. We have now indicated the number of replicates in the material and methods section ( line 281 in Statistical Analysis).

-For both tables and figures, the legends should more completely describe the treatments so that the reader does not have to refer back to the text to understand tables and figures independently.

More information about the treatments is now included in the tables. The treatment it figures legends was described. 

Figure 2 legend: This sentence makes no sense: “High molecular weight proteins are represented above in the figure, while low molecular weight proteins are located below.” Please explain or delete.

Sorry, it is a mistake. This sentence has been deleted.

Reviewer 2 Report

The seasonal reproduction of sheep is regulated by changes in photoperiod which mediated by nocturnal melatonin secretion. In this study, authors explored the effects of melatonin on ram reproduction in equatorial photoperiod. Numerous studies have also demonstrated that Melatonin can prevent sperm from ROS or oxidative stress. This manuscript was soundly designed and well written, and the results were also well presented. I suggest this manuscript can be accepted for publication after minor revision.

Minor comments:

  1. The situation that sperm was capacitated or not has nothing to do with the dry or rainy period of the sheep. So, I think the title of this manuscript should be revised.
  2. In conclusion section, it was not suitable for authors concluded that melatonin has no role in seasonal control of reproduction.
  3. The indicators of significance in figure 1 and 3 were not clearly presented and hard to understand.

Author Response

Thank you for your comments and suggestions.

1.The situation that sperm was capacitated or not has nothing to do with the dry or rainy period of the sheep. So, I think the title of this manuscript should be revised.

The aim of the manuscript was to evaluate the response to in vitro capacitation in spermatozoa obtained from three sheep breeds reared in Colombia under a photoperiodic regimen of 12L:12D. The second objective was to elucidate whether melatonin can regulate ram sperm capacitation in these breeds in a medium with cAMP elevating agents. Both evaluations were conducted during the rainy and dry seasons because maybe environmental conditions could affect sperm behavior and response. In our opinion the tittle fits the aim and the results of the study, but perhaps we have not understood the reviewer's concern.

2. In conclusion section, it was not suitable for authors concluded that melatonin has no role in seasonal control of reproduction.

The reviewer is right. The sentence has been changed by: “This study shows that melatonin is able to exert direct effects on spermatozoa in ovine breeds located under equatorial photoperiod, as is the case with seasonal breeds located in temperate regions” (line  42 and 589).

3. The indicators of significance in figure 1 and 3 were not clearly presented and hard to understand.

There were some mistakes and missing information in the Figure 1 and 2 legends that we have now mended. We have used the same nomenclature than in the tables. So, we use:

  • *: for representing significant differences between seasons within the same treatment and breed
  • different capital letters: for significant differences between treatments (swim-up, Cap-TALP, and Cap-CK) within breed and season
  • different numbers: for differences between treatments (Cap-CK, Cap-CK-100pM MEL, and Cap-CK-1µM MEL) within breed and season. Here, we use different nomenclature than capital letters because with capital letters we try to show significant differences when the effect of the capacitation media was studied whereas with numbers, we try to show significant differences when the effect of the inclusion of melatonin in the cocktail media was analyzed. If we had used the same nomenclature for all experimental groups, there are some significant differences between all groups (for instance Cap-CK MEL vs. swim-up) that are not relevant for the study, and that would complicate the understanding. In fact, we have previously tried to use capital letters for significant differences between all treatments, but there were so many letters on some groups that it was difficult to know with which one there were differences.
  • And finally different lowercase letters for significant differences between breeds within treatment and season. This information was missed and it has now been added to the legend.